# Green Innovation and Synthesis of Honeybee Products-Mediated Nanoparticles: Potential Approaches and Wide Applications

**DOI:** 10.3390/bioengineering11080829

**Published:** 2024-08-14

**Authors:** Shaden A. M. Khalifa, Aya A. Shetaia, Nehal Eid, Aida A. Abd El-Wahed, Tariq Z. Abolibda, Abdelfatteh El Omri, Qiang Yu, Mohamed A. Shenashen, Hidayat Hussain, Mohamed F. Salem, Zhiming Guo, Abdulaziz M. Alanazi, Hesham R. El-Seedi

**Affiliations:** 1International Research Center for Food Nutrition and Safety, Jiangsu University, Zhenjiang 212013, China; shaden.khalifa@regionstockholm.se; 2Neurology and Psychiatry Department, Capio Saint Göran’s Hospital, Sankt Göransplan 1, 112 19 Stockholm, Sweden; 3Department of Chemistry, Faculty of Science, Menoufia University, Shebin El-Kom 31100107, Egypt; aya.shetaia@gmail.com (A.A.S.); nehaleid45@gmail.com (N.E.); 4Department of Bee Research, Plant Protection Research Institute, Agricultural Research Centre, Giza 12627, Egypt; aidaabd.elwahed@arc.sci.eg; 5Department of Chemistry, Faculty of Science, Islamic University of Madinah, Madinah 42351, Saudi Arabia; mashenashen@gmail.com (M.A.S.); a-aziz@iu.edu.sa (A.M.A.); 6Surgical Research Section, Department of Surgery, Hamad Medical Corporation, Doha 3050, Qatar; aelomri@hamad.qa; 7Vice President for Medical and Health Sciences Office, QU-Health, Qatar University, Doha P.O. Box 2713, Qatar; 8Polysaccharides of Jiangxi Province, Nanchang University College of Food Science and Technology, 235 Nanjing East Road, Nanchang 330047, China; yuqiang8612@163.com; 9National Institute for Materials Science (NIMS), 1-2-1 Sengen, Tsukuba-Shi 305-0047, Ibaraki-Ken, Japan; 10Department of Bioorganic Chemistry, Leibniz Institute of Plant Biochemistry, Weinberg 3, 06120 Halle (Saale), Germany; hidayat.hussain@ipb-halle.de; 11Department of Environmental Biotechnology, Genetic Engineering and Biotechnology Research Institute, GEBRI, University of Sadat City, Sadat City P.O. Box 79, Egypt; mohamed.salem@gebri.usc.edu.eg; 12School of Food and Biological Engineering, Jiangsu University, Zhenjiang 212013, China; guozhiming@ujs.edu.cn

**Keywords:** bee products, nanoparticles, biological activities, catalytic application, food industries

## Abstract

Bee products, abundant in bioactive ingredients, have been utilized in both traditional and contemporary medicine. Their antioxidant, antimicrobial, and anti-inflammatory properties make them valuable for food, preservation, and cosmetics applications. Honeybees are a vast reservoir of potentially beneficial products such as honey, bee pollen, bee bread, beeswax, bee venom, and royal jelly. These products are rich in metabolites vital to human health, including proteins, amino acids, peptides, enzymes, sugars, vitamins, polyphenols, flavonoids, and minerals. The advancement of nanotechnology has led to a continuous search for new natural sources that can facilitate the easy, low-cost, and eco-friendly synthesis of nanomaterials. Nanoparticles (NPs) are actively synthesized using honeybee products, which serve dual purposes in preventive and interceptive treatment strategies due to their richness in essential metabolites. This review aims to highlight the potential role of bee products in this line and their applications as catalysts and food preservatives and to point out their anticancer, antibacterial, antifungal, and antioxidant underlying impacts. The research used several online databases, namely Google Scholar, Science Direct, and Sci Finder. The overall findings suggest that these bee-derived substances exhibit remarkable properties, making them promising candidates for the economical and eco-friendly production of NPs.

## 1. Introduction

Nanotechnology is an interdisciplinary science that explores the unique features of materials and systems at the nanoscale and occupies an important position at the cutting edge of scientific innovation and research. The combination of pharmaceutical science and nanotechnology has enabled the precise design of nanoscaled drug delivery systems, facilitating targeted drug administration, improving bioavailability, and reducing side effects compared to original materials. Nanomaterials can penetrate biological barriers, target specific tissues or cells, and enhance therapeutic outcomes [1]. Nanoparticles’ (NPs) tiny sizes are the secret behind their unique physicochemical properties such as the control of the surface area, which is a contributory factor to strong surface reactivity [2]. Therefore, particles with diameters below 100 nm have superior features relative to their corresponding bulk materials [3], making them potential candidates in preclinical and clinical trials. Unfortunately, NPs are synthesized via chemical and physical methods by laser ablation co-precipitation, pulsed wire discharge, vacuum vapor deposition, high-energy irradiation, lithography, and mechanical milling [4], usually requiring need special conditions. Moreover, they are costly and non-eco-friendly due to the use of toxic and hazardous chemicals [5,6]. In contrast, biosynthetic processes that depend on green chemistry-based methods present simple, nontoxic, and eco-friendly protocols at room temperature and pressure [7] and exhibit a wide range of NP sizes, physicochemical properties, shapes, and compositions [8].

Green synthesis of nanomaterials aims to develop and design nanomaterials utilizing sustainable techniques and renewable resources, such as biological agents or plant extracts, thus reducing the environmental impact of conventional chemical processes and producing materials with unique properties and applications in the sectors of healthcare, catalysis, and environmental remediation [9]. This strategy contributes to the development of cleaner and more efficient nanotechnology-based solutions, increasing the likelihood of a greener and more responsible future in materials science and technology [10]. Therefore, scientists have been paying great attention to exploring new natural materials in the green synthesis of NPs. Natural products such as plants, microorganisms like algae, fungi, marine, and bee products are highly recommended for NP synthesis. Those natural products are an enormous treasury of bioactive chemical compounds, classified as proteins, peptides, flavonoids, phenolic compounds, terpenes, minerals, sugars, and fatty acids [11,12]. Biogenic synthesis is a “Bottom Up’’ approach that is similar to chemical reduction but substitutes the reducing agent with natural extracts that possess stabilizing, growth terminating, and capping properties [10].

Honeybee products are important natural sources that include several substances such as honey, propolis, beeswax, royal jelly, bee pollen, bee bread, and bee venom. These products have been known for their medicinal and health-promoting properties since ancient times [12]. Nowadays, they have reached preclinical and clinical stages owing to their antioxidant, antimicrobial, anti-inflammatory, and neuroprotective impacts [13].

Bee products are highly effective in the green synthesis of NPs due to their unique components that serve as reducing and capping agents, i.e., carbohydrates, flavonoids, and proteins. In addition, the simple, cost-effective, and eco-friendly process of green synthesis using honeybee products for NPs production provides many advantages for medical and pharmaceutical applications [14,15]. This study aims to provide an updated overview of this subject, focusing specifically on the role of bee products in the green synthesis of NPs, highlighting the biological properties and biomedical applications. This research is a part of our continuous exploration of the green synthesis of NPs [16,17,18].

## 2. Method of Search

In this regard, original articles within the field of green and eco-friendly synthesis of NPs employing bee-derived products have been compiled. Our research involved the utilization of online databases, including Google Scholar, Science Direct, and Sci Finder, to support our investigation [19]. This study examined the pertinent bee products employed in the green and eco-friendly synthesis of NPs. It delved into the essential components of these bee products, which serve as both reducing and capping agents. Moreover, it explored the properties of the NPs produced, the methodologies utilized for their characterization, and their biological activities and applications. Numerous published studies were systematically reviewed by scrutinizing articles from databases to identify the most relevant work, and only those aligning with the review objective were included in our review citations. As a result, only 68 research articles on the use of bee products in the green synthesis of NPs are cited in this review. The strategy’s flowchart is illustrated in Figure 1.

## 3. Bee Products and Their Metabolites

### 3.1. Honey

Honey has been known as a valuable, nutritive, and healthy food since the distant past. Honey production is a secondary consequence of honeybee actions related to plant nectar, secretions, and excretions [20,21]. In general, honey is a concentrated solution that contains a large amount of carbohydrates (80–85%), proteins (0.1–0.4%), and water (17–20%), as well as amino acids, vitamins, enzymes, antioxidants, and antibiotics [22]. Honey can also be used for therapeutic purposes against eye, skin, ear, stomach, and intestinal diseases. Based on modern medical literature, honey can aid in the preventive and treatment strategies against several diseases such as obesity, diabetes, metabolic syndrome, and cancer [23]. Furthermore, honey protects the nerve cells against damage by reducing and preventing inflammation [24] and due to its antioxidant and antimicrobial properties [25].

Monosaccharides, glucose, and fructose make up up to 95% of honey’s dry weight [13]. The chief role of glucose and fructose is claimed to be their reduction properties [26,27,28] (Figure 2). The two sugars act as reducing sugars; they react with Tollens’s (reduction of Ag+ to Ag0), Benedict’s (Cu^2+^ to Cu_2_O), and Fehling’s reagents [29]. These chemical properties make honey a good choice as a green reducing and capping agent in NPs synthesis. Although it has been used for many years, its use has been constantly increasing since 2009. Additionally, the pH value in all honeys ranges from 3.5 and 5.5 due to their contents of organic acids, especially gluconic acid [30].

Thus, honey can act as a stabilizing and reducing agent, at room conditions and without any other chemicals. Thus, this method is considered safe, rapid, low-cost, reproducible, and simple as well as effective in many applications [31]. Honey has a role in the synthesis of silver NPs (AgNPs), platinum NPs (PtNPs), zinc oxide NPs (ZnONPs), calcium oxide NPs (CaONPs), decahedral cinnamon NPs(DCNPs), cerium oxide NPs (CeO_2_NPs), cobalt ferrite NPs (CoFe_2_O_4_NPs), Ag dopedNPs, spinel ferrite NPs (NiFe_2_O_4_NPs), graphene basedNiO_2_/Cu_2_O nanocomposite (Gr@NiO_2_/Cu_2_O NCs), chromium oxide NPs (Cr_2_O_3_-NPs), and copper NPs (CuNPs) [31].

### 3.2. Propolis

Propolis (bee glue) is the resinous material produced by honeybees when mixing collected tree buds of different plant sources with bee saliva, pollen, and wax. The propolis color varies according to flora, and it can be brown, green, red, and black [32]. Propolis is an unrivaled product with a complex composition that includes more than 420 chemical substances [33]. Propolis constituents include resins (50%), bee pollen (5%), beeswax (30%), and aromatic and essential oils (10%). Propolis contains flavonoids, amino acids, vitamins, polyphenols, and micronutrients. Cinnamic acid, or cinnamyl alcohol, terpenes and sesquiterpene-alcohols and their derivatives, benzoic acid and its derivatives, sterols or steroid hydrocarbons, and sugars are also some of the isolated components [16]. Several organic compounds such as galangin, vanillin, isalpinin, chrysin, pinocembrin, caffeic acid, phenethyl ester, tetochrysin, phenethyl ester, and ferulic acid have been identified [34]. Recently, many drugs targeting studies have focused on the propolis bioactive compounds due to their great therapeutic activities [35]. Propolis has been shown to have a positive effect against COVID-19 [36], where in an in vitro study, the antiviral effects of propolis extracts from Mexico and Brazil and their phenolic compounds (quercetin, caffeic acid, and rutin) were tested against the infection of human lung fibroblast cells (MRC-5) by human coronavirus (HCoV-229E). All samples showed antiviral activity; however, the quercetin showed the best EC_50_ value of 77.208 µg/mL [36]. Similar to honey [37,38], propolis has been involved in the green synthesis of selenium NPs (SeNPs) [39], AgNPs [40], Ag–PtNPs, palladium NPS (PdNPs) [41], gold NPs (AuNPs) [42], CuNPs [43], Au@AgNPs [44], iron oxide NPs (IONPs) [45], titanium oxide NPs (TiO_2_NPs) [46], copper oxide NPs (CuONPs) [47], ZnONPs [48], AuNPs@ZnO [48], and AgNPs@ZnO [48].

With its unique chemical composition of flavonoids, polyphenols, and other bioactive compounds, propolis has emerged as a highly intriguing candidate for nanomaterials’ synthesis since 2009. This composition offers a flexible platform for designing nanomaterials with specific features, making it a fascinating subject for materials science and nanotechnology research. Moreover, propolis is a naturally occurring material that is sustainable and gives an eco-friendly twist to the creation of advanced nanomaterials [40,49].

### 3.3. Bee Pollen

Bee pollen (BP) is a product of flowering plants and plays a chief role in the honeybee’s nutrition. It is very important for the development of muscles, broods, and digestive systems of young bees and royal jelly production due to its high content of proteins [50]. BP is thought to possess a variety of health-promoting features in addition to its nutritional benefits. Some proponents claim that it can boost energy, improve immune function, and alleviate allergy symptoms. Although there is still a lack of scientific evidence to support these claims, its antioxidant properties are well-documented, where it helps neutralize free radicals in the body, potentially reducing the risk of chronic diseases and supporting overall health [51].

Consequently, BP has gained popularity in health-conscious circles as a natural supplement and superfood owing to its enrichment of vitamins, minerals, proteins, and enzymes. BP contains proteins, carbohydrates, lipids, vitamins, amino acids, and high amounts of polyphenolic flavonoids. Several biological activities of BP, such as its anti-inflammatory, antioxidant, anticancer, antifungal, and antimicrobial properties, have been reported. Furthermore, its curative properties against anemia, allergy, atherosclerosis, prostatic enlargement, osteoporosis and function in liver protection have also been documented [52].

The diversity of BP chemistry aids the synthesis of Ag, Mg, and Au NPs. BP and its metabolites act as powerful reducing and capping agents [53,54,55]. AgNPs exhibit various preferable characterizations related to their carbon, oxygen, and nitrogen elements, in addition to their biological activities [53,54,55].

The intricate and naturally existing microstructures within BP offer an optimal template for nanomaterial design with specific features. Through the utilization of the inherent structures found in bee pollen, researchers aim to develop nanomaterial applications in diverse sectors, including medicine, electronics, and environmental remediation [55].

### 3.4. Bee Bread

Honeybees gather nectar and pollen from flora, which they store in honeycombs. Here, the collected substances undergo fermentation due to the action of lactic acid bacteria and digestive enzymes, resulting in a concoction known as Bee bread (Bb). This mixture has an acidic pH ranging from 3.5 to 4.2. Bb serves as the primary protein source in the beehive, providing nourishment for both larvae and adult bees [56]. Bb is a rich source of various chemical constituents. It contains proteins, amino acids, and carbohydrates, along with aldehydes, unsaturated aliphatic acids, alcohols, ketones, and nitriles. Additionally, it is packed with carotenoids, terpene derivatives, polyphenols, and flavonoids. The presence of vitamins such as vitamin C, B, K, P, and E, as well as furfural, further enhances its nutritional profile [56].

Numerous studies have highlighted the diverse biological activities of Bb, including its antimicrobial, anti-inflammatory, antioxidant, and anticancer properties. Rutin, a significant phenolic compound, is found in Bb, and its high content serves as an indicator of the bread’s freshness [57]. Recently, it has gained popularity as a dietary supplement, with a recommended intake similar to that of BP, ranging from 20 to 40 g/dalton [52]. The components of Bb not only exhibit unique biological activities but also possess potential reducing powers. These effects contribute to its success as a natural product used in the eco-friendly synthesis of NPs (Figure 2) [56].

### 3.5. Royal Jelly

Royal jelly (RJ) is one of the bee products that are widely used in functional food, in cosmetics, and for the treatment of many illnesses. It is a creamy acidic secretion from the mandibular and hypo-pharyngeal glands of worker bees [58]. In addition, it is regarded as the only food given to bee larvae for 1–3 days and to the queen for her life period, and it helps in the regulation of honeybee physiological function due to its advanced chemical profiling [19]. RJ contains water (60–70%), total sugar (10–16%), crude protein (12–15%), lipids (3–6%), vitamins, free amino acids, phenols, flavonoids, and salts [58,59]. RJ biological activities have been studied in vitro and in vivo and have also been approached in many clinical studies. The effects of RJ on immune and memory regulation, digestive system enhancement, glucose level regulation, reduction of obesity, and as anti-inflammatory and anticancer entities have been proved. In addition, RJ showed high ability in the green synthesis of AgNPs where spherical, nanodisc, and clustered NPs were seen. In addition, they showed antibacterial activity against both Gram-positive and Gram-negative bacteria [60,61].

### 3.6. Beeswax

Beeswax is the substance that bees use for building their honeycombs and is where they store the honey and Bb. It has interesting hydroprotective effects explaining its appeal in cosmetics, body products, and the food industry [62]. Recently, many of the various biological activities of beeswax have been demonstrated. It can be used in healing bruises, burns, and inflammation and has also demonstrated antimicrobial activity against *Candida albicans*, *Aspergillus niger*, *Staphylococcus aureus*, and *Salmonella enterica* [62]. The majority of beeswax chemical constituents (>50), i.e., odd and even hydrocarbons, palmitate, hydroxypalmitate monoesters, oleate, and olefin, have been detected. Moreover, palmitate and oleate monoesters esterified with 1-octadecanol and 1eicosanol were detected in beeswax for the first time [63]. Beeswax has been used as a coating agent for propolis NPs because of its interesting physical and chemical features. It is used in the encapsulation process of propolis NPs to protect them from environmental conditions [64]. Despite that, beeswax has not been used in the preparation of metallic NPs [63].

### 3.7. Bee Venom

Bee venom (BV) is a vital product produced by *Apis mellifera* anatoliaca worker bees, where it is formed in the acid gland (venom gland). BV plays a defensive role and has gotten a lot of attention owing to its bioactive constituents, e.g., peptides, proteins, enzymes, volatile compounds, and carbohydrates. BV contains melittin, adolapin, apamin, tertiapin, secapin, mast cell-degranulating phospholipase A2, histamine, epinephrine, and hyaluronidase, which are types of peptides and free amino acids [65]. Melittin is the main component of BV. It occupies 40–50% of its dry weight. Melittin is a polypeptide with 26 linear amino acid sequences. It is characterized by cationic and hemolytic as well as amphipathic properties [58].

BV is well known for its anti-inflammatory, antiviral, anticancer, antibacterial, antiarthritis, and antifungal metabolic effects. It also has a great impact on the immune system, cardiovascular system, and central and peripheral nervous systems. Melittin has similar biological activities, as reported by [65].

Interestingly, wasp venom, which has a similar chemical profile to BV, was used in the green synthesis of AuNPs, as reported recently by [66]. BV nanoconjugates showed significant anticancer activity [67], whereas BV conjugated with nano chitosan from shrimp exoskeletons was proven effective against prostate cancer (PC3) and hepatocellular carcinoma (HEPG2) cell lines [68]. 

## 4. Green Synthesis of NPs Utilizing Bee Products

Honeybees’ products, including honey, propolis, RJ, BP, Bb, BV, and beeswax, are rich in various bioactive compounds, i.e., protein, peptides, minerals, terpenes, phenolic, and flavonoid compounds, explaining their wide range of biological activities [13]. In the last decades, NPs’ green synthesis using bee products has attracted more attention [52]. The technique was described as adding an aqueous extract of a bee product (in the case of BP) to enhance the reduction and capping processes, where polyphenols and flavonoid compounds were suggested to be the key players (Figure 2) [55]. Later, the same protocol was applied to prepare AgNPs [69], AuNPs [70], solid lipid NPs, liposomes, magnetic NPs, mesoporous silica NPs, and liquid crystalline formulations [71]. The spherically shaped, highly crystalline formed AgNPs showed antimicrobial and anticancer activity against both human breast cancer cell line (MCF-7) and liver carcinoma (HepG2)cell lines, respectively [72].

### 4.1. Sliver Nanoparticles (AgNPs)

AgNPs have captured attention in recent years as the most eminent NPs due to their special properties and varied applications. In particular, AgNPs have positive effects on agriculture and plant biotechnology, as they improve the seed germination and plant growth and enhance the photosynthetic process. Furthermore, they are considered safe pesticides and vital fertilizers [73]. The AgNPs’ distinct biological activities, including their antiviral, antifungal, and antibacterial impacts, allow them to contribute to food services, healthcare, medical devices, building materials, household materials, and cosmetics industries [74].

#### 4.1.1. Utilizing Honey for AgNP Synthesis

Honey plays a crucial role in nanoparticle synthesis, serving as a reducing and capping agent for metals. Honey has been utilized in the synthesis of AgNPs, which are characterized by a size range of 5–98 nm and a spherical, crystalline shape, as detailed in Table 1, Figure 3.

Natural honey from Kerala was employed as a reducing agent to produce tiny AgNPs, approximately 4 nm in size, using a straightforward and environmentally friendly method. This process was conducted in water at room temperature. Specifically, 15 mL of honey was added to 20 mL of an aqueous solution of AgNO_3_ (10^−3^ M), and the mixture was stirred thoroughly for 1 min. It was suggested that glucose served as the reducing agent, while proteins were likely the capping materials [92]. In 2013, a method was developed that utilized natural honey and sunlight irradiation to simplify and expedite the synthesis of AgNO_3_. In this process, honey eliminates the need for any stabilizing intermediates, serving as both the reducing agent and the capping agent [80]. Specifically, 5 mL of Nigerian honey was combined with 95 mL of 0.1 M AgNO_3_ aqueous solution and thoroughly mixed. Within seconds, the solution turned to a yellowish-brown color, signaling the formation of a silver colloid. These AgNPs demonstrated remarkable stability, maintaining their integrity for over six months and offering substantial resistance to erosion in an acidic solution [80]. In 2011, Sreelakshmi and her team introduced a novel strategy for stabilizing AuNPs and AgNPs. This strategy utilized natural honey and was executed in a single step at room temperature. The honey polysaccharides were instrumental in the reduction and stabilization of the metallic ions [70].

#### 4.1.2. Utilizing Propolis for AgNPs’ Synthesis 

The aqueous extract of Egyptian propolis has been used as a reducing and capping agent in the green synthesis of AgNPs applying a simple, environmentally friendly method [37]. The aqueous AgNO_3_ was reduced to produce AgNPs with a diameter between 9.43 nm to 13.09 nm. The speed of reaction was 13 min at the temperature of 75 °C. The product has mostly sphere-like shapes as confirmed by scanning electron microscopy (SEM) and transmission electron microscopy (TEM) analysis, and a crystalline form was seen by the X-ray diffraction (XRD) pattern. The study also demonstrated the antibacterial role of the biosynthesized AgNPs against *Escherichia coli* [37,93].

#### 4.1.3. Utilizing BP for AgNPs’ Synthesis 

To benefit from the bee pollen’s active ingredients, some studies have used the product in the green synthesis of metallic NPs [55,94]. Turunc used *Cupressus sempervirens* L. (CSPE) pollen extract in green synthesis of AgNPs via a simple, cost-saving, and environmentally friendly technique. CSPE was used as a reducing and stabilizing agent. AgNPs formed and were characterized by an absorption peak at 440 nm. Flavonoids, alkaloids, tannins, steroids, etc., are thought to be responsible for the reduction of Ag+ to Ag0. The formed AgNPs were seen as spherical by SEM and TEM images. The results of energy-dispersive X-ray (EDX) analysis indicated the occurrence of the elements carbon, oxygen, and nitrogen. XRD data have determined the size of crystallite AgNPs to be 11.55 nm. The study also demonstrated the significant antioxidant performance of the biosynthesized AgNPs [55]. In another study, AgNPs exhibited antioxidant ability and anticancer activity against both MCF-7 and HepG2 cell lines [72]. BP from Amasya, Turkey, was administrated for bio synthesizing of silver nanoparticles (BP-AgNPs). The average sizes of the obtained spherical-shaped particles were 40–60 nm. The particles were identified using spectroscopic and microscopic techniques. The total phenolic content was 12.07 ± 0.03 mg GAE/mL and has significant inhibition against α-amylase and α-glucosidase, diabetes mellitus-related enzymes at IC_50_ values of 2.56 ± 0.10 and 2.13 ± 0.11 mg/mL, respectively, and thus the study highly recommended BP-AgNPs for the treatment of diabetes mellitus [94]. The BP components, especially phenolic compounds, have a major and powerful reducing property [94]. The hydrophobic nature of propolis often besets its use due to these biopharmaceutical issues, and thus to improve and develop its bioavailability, many studies have taken the approach of developing nano-propolis, as mentioned in Table 1 [95]. AgNPs have been synthesized using BP, and after analyzing the results by analytical techniques, the biosynthesized AgNPs were shown to be spherical in shape with a diameter of 10–35 nm. The obtained AgNPs remained stable for a month under the lab conditions. Furthermore, the antioxidant properties of AgNPs were investigated using 1,1-diphenyl-2-picrylhydrazyl (DPPH). The study recommended the large-scale production of propolis-mediated NPs without the need to use any chemical solvents [96].

#### 4.1.4. Utilizing Bee Bread for AgNPs’ Synthesis

Bb extracts from the *Apis mellifera* L. colonies in Romania were used as reducing and stabilizing agents in the biosynthesis of AgNPs prior to the characterization using ultraviolet–visible (UV-Vis) spectroscopy, zeta potential, and TEM, and displayed a size of 48.3 to 150.1 nm [91]. The study suggests that the abundance of various polyphenolic substances, with kaempferol being a key component, influenced the size and form of the particles generated. The antioxidant activity of the AgNPs was determined utilizing DPPH, 2,2′-azino-bis(3-ethylbenzothiazoline-6-sulfonic acid (ABTS), and fluorescence recovery after photobleaching (FRAP) assays. In addition to the antimicrobial activity against the Gram-positive bacteria *E. faecalis*, *S. aureus B. cereus*, and the Gram-negative bacteria *E. coli*, *P. aeruginosa*, *S. enteritidis*, and the yeast *C. albicans*, the anti-proliferative effect against the human colon adenocarcinoma cell line was reported [91].

#### 4.1.5. Utilizing Royal Jelly for AgNPs Synthesis 

Northeastern Mexican RJ was utilized to bio-synthesize AgNPs from aqueous AgNO_3_ solutions without the utilization of additional stabilizing agents or sources of heating to start the oxidation-reduction reactions [60]. The majority of the produced NPs were ultra-fine Ag single crystals with disk-like shapes and diameters of about 4 nm. Additionally, 4, 4′-dimethyldiazoaminobenzene nano-disks of silver and twined fine Ag particles were coated by RJ using the protein and carbohydrate residue [60]. Armenian RJ was used as a reducing and oxidizing agent in the environmentally friendly process of AgNPs production. The 430 nm AgNPs were seen by UV-Vis absorption, and the peak fluorescence emission was corresponded to 487 nm, while the spherical shape was determined using TEM and SEM. Atomic force microscopy (AFM) was also utilized to analyze the effect of ultrasound treatment on the AgNPs. Statistical analysis showed that the sizes of AgNPs were 7.8–192.7 nm and 8.6–61.8 nm, before and after ultrasound treatment, respectively. The treatment with ultrasound helped to decrease the size of AgNPs and increase the homogeneity of the distribution, which led to higher activity against *Salmonella typhimurium* than *S. aureus*, as shown in Figure 4 [61].

Gevorgyan et al. [97] determined the chemical structure and antibacterial activity of synthesized RJ-AgNPs. The eco-friendly synthesis of AgNPs using a low-molecular-weight fraction extracted from RJ provides insights into nanoparticle characterization and emphasizes their antibacterial properties. Spherical and clustered NPs were noticed by TEM, and the crystallinity was affirmed by selected area electron diffraction patterns (SAED) and XRD. The presence of silver oxide and organic materials was further supported by XRD. The AgNPs’ sizes were around 50–100 nm, but the size increased parallel to the increase in the concentration of silver ion precursor [97]. The provided data and findings affirm that the low-molecular-weight fraction extracted from RJ can effectively facilitate the green synthesis of AgNPs. Green-synthesized AgNPs exhibited antibacterial effects against Gram-positive and Gram-negative bacteria, where Gram-negative bacteria have greater susceptibility to all tested NPs [97]. 

Therefore, approaching RJ for nano biosynthesis purposes is highly recommended due to its high antioxidant potential. RJ can be used in the green synthesis of NPs as a reducing and stabilizing agent at room temperature, and, moreover, it can be economically reasonable and helps decrease the risks to the environment and human health, as described in Figure 5.

### 4.2. Gold Nanoparticles (AuNPs)

AuNPs are among the most essential NPs and have been widely used for medical and non-medical applications as ideal materials because of their unique distinct characteristics, e.g., inert, biocompatible, and low toxicity. AuNPs have diagnostic and therapeutic uses, including biosensor applications, anticancer applications, and drug delivery of nano-vehicles to targeted tissues [99,100].

#### 4.2.1. Utilizing Honey for Gold Nanoparticles’ (AuNPs) Synthesis 

An amount of 10 g of honey was diluted in 100 mL of water and then added to 100 mL of an aqueous HAuCl_4_ (10^−3^ M) solution based on a previous method [92]. This mixture was left at room temperature for 2 h to facilitate the reduction of Au^3+^ [70]. Transmission electron microscopy (TEM) images revealed that the average size of the nanoparticles was approximately 9.9 nm. These NPs exhibited a higher level of dispersion compared to AuNPs, which could be attributed to the inherent properties of the metals, such as melting point and surface energy. Notably, no agglomeration of Ag and Au was observed, which was attributed to the honey functional groups that served to anchor the NPs [70]. In 2013, Haiza and her team utilized natural honey from Malaysia, specifically Tualang honey, in the stabilization and reduction process. This was chosen based on the honey’s health benefits and environmentally friendly properties and as an alternative to harsh compounds such as sodium borohydride and dimethyl formamide [101].

#### 4.2.2. Utilizing Propolis for Gold Nanoparticles’ (AuNPs) Synthesis 

Propolis has been used to produce AuNPs due to its high content of flavonoids, terpenoids, and polyphenolic acids that act as a reducing agent for Au^+3^ to Au (Table 2 and Figure 6) [42].
bioengineering-11-00829-t002_Table 2Table 2Synthesis of nanoparticles bio fabricated using bee products.NanoparticlesBee Products/Reducing and Capping AgentsMorphologyNPs Size (nm) Biological Activities and ApplicationReferencesGold nanoparticles (AuNPs)Propolis/Carboxyl groups in the propolis component Spherical7.8Anticancer activity[99](Au@AgNPs)Propolis/Polyphenolic moleculesSpherical108Antibacterial and anticancer activity[44]AuNPsPropolis/Polyphenols, proteins, vitamins, and sugarsSpherical, hexagonal, and triangular20–60Catalytic and anti-tumor activity[42]AuNPs@ZnOPropolisSpherical22–73Antibacterial activity and cytotoxicity properties against human breast cancer cell line (MCF-7) and liver carcinoma cell line (HepG2). [48]Palladium nanoparticles (PdNPs)Propolis/FlavonoidCrystalline phase and face-centered cubic structure3.14–4.62 Anticancer and antimicrobial[41]Selenium nanoparticles (SeNPs)Propolis/Phenolic content Spherical279Antibacterial activity[49]SeNPsPropolis/Flavones, steroids, phosphoric acid, charcones, acetic acid, butanol, butyl ester, butanoic acid, hydroxyl and keto waxes, ketones, terpenoids, and sugarsSpherical50–60Antioxidant activity[102]SeNPsPropolis/Alcohol and polyphenolsCrystalline, oval, and with smooth surface 52.9–118Antioxidant and antimicrobial potential[103]Copper nanoparticles (CuNPs)Propolis/Proteins,sugars, and polyphenolsSpherical15–23N.D.[43]CuO NPsPropolisCrystallite75–145 and 120–155Antimicrobial, and cytotoxic effects[47]Iron oxide nanoparticles (IONPs)PropolisSphericalAround 87 and 194 Antibacterial activity and degradation of dyes[45]Titanium oxide nanoparticles (TiO_2_NPs)flavonoids and phenolic compounds Quasi-spherical21Antimicrobial activity[46]calcium oxide nanoparticles (CaONPs)Honey/Hydroxyl and carboxylic groups, amines, and amidesSpherical >100Antifungal activity and cytotoxicity properties[104]Decahedral cinnamon nanoparticlesHoney/Vitamins, proteins, monosaccharide, and fructoseCrystalline17.08 ± 4.71Antibacterial activity[105]Cerium oxide anoparticles (CeO_2_NPs) Honey/Proteins and carbohydratesSpherical1.23Antioxidant and photocatalytic dye degradation [106]CeO_2_NPs Honey/Proteins and carbohydratesSpherical2.61Antioxidant and photocatalytic dye degradation [106]CeO_2_NPs Honey/Proteins and carbohydratesSpherical3.02Antioxidant and photocatalytic dye degradation [106]Graphene basedNiO_2_/Cu_2_O nanocomposite (Gr@NiO_2_/Cu_2_O NCs)Honey/Reducing sugarCrystalline 15 Catalyst in synthesizing the functionalized Schiff-base derivatives[107]Chromium oxide nanoparticles(Cr_2_O_3_NPs)Honey/Proteins, and carbohydratesCrystalline20Antioxidant and anti-inflammatory activity[108]Cr_2_O_3_NPs Honey/CarbohydratesCrystalline24.7205 Antioxidant and antibacterial activity[109]CoFe_2_O_4_, Ag dopedNPsHoneyCrystallite24–41Antibacterial activity[110]Platinum nanoparticles (PtNPs)Honey/ProteinCrystalline5–15Catalytic application on preparation of organic dye[8]Zinc oxide nanoparticles (ZnONPs)Honey/Fructose, glucose, sucrose, proteins, minerals, and vitamins.Crystalline39Catalytic, antibacterial, and antifungal activities[111]ZnONPs PropolisSpherical17–70Antibacterial activity, cytotoxicity properties against cell lines (MCF-7 and HepG2). [48]NiFe_2_O_4_ spinel ferriteNPsHoney/Glucose and fructoseOctahedral10–70Magnetic, dielectric, and electrical properties[112]Carbon Nanoparticles (CNPs)Honey/Monosaccharides and the higher sugarsSphericalBelow 7-[113]AgNPs@ZnOPropolisSpherical45–75Antibacterial activity, cytotoxicity properties against cell lines (MCF-7 and HepG2). [48]BP@MgNPsBee pollenSpherical36–40Antioxidant and antibacterial activity[53]
Figure 6(**A**) The overall diagrammatic preparation of CuNPs derived from Honey, and (**B**) illustrates the anti-cancer properties of the synthesized CuNPs (Figure used with permission) [114].
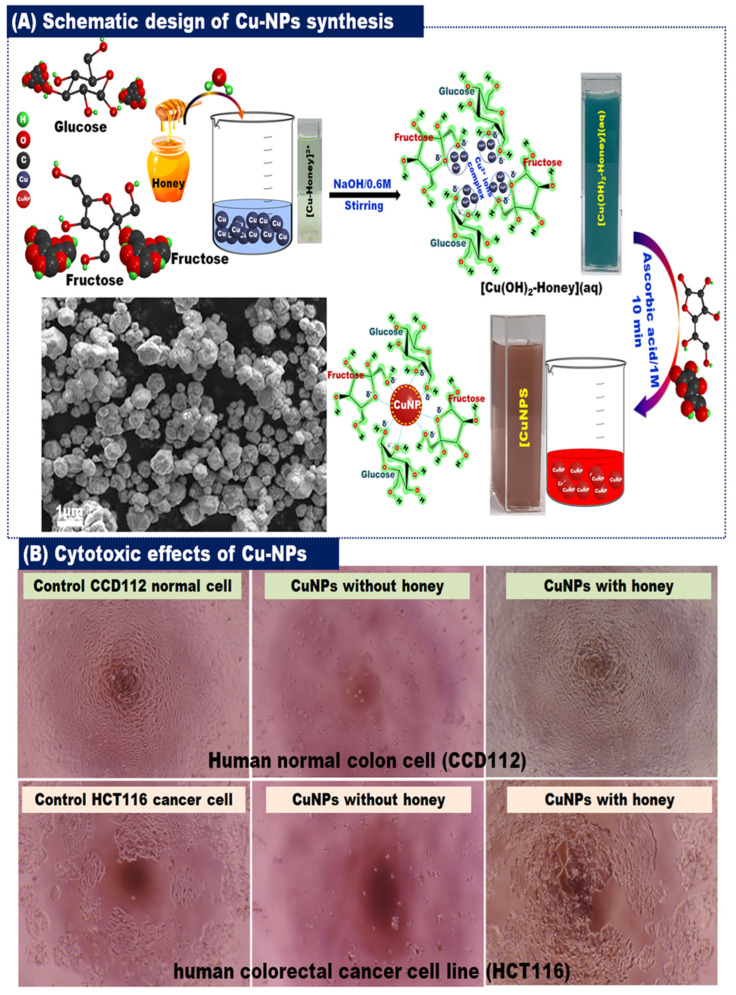



The ethanol and water extracts of Indian propolis and two biochemical constituents isolated from it, namely pinocembrin and galangin, have been used in the synthesis of Ag and Au nanoparticles. TEM images were used to detect the size and morphology of the synthesized NPs. The results showed a wide spectrum of morphologies, spheroidal shape with many nano-rods, nano-wedges, and nano-prisms of several regular geometrical shapes, such as squares, triangles, hexagons, and trapezoids. The particles formed by pinocembrin and galangin have a more homogeneous size than the parent materials, with a size of 20 to 50 nm. Taken together, propolis and its constituents showed equal efficiency in the rapid synthesis and excellent stabilization of the resultant particles [38].

Botteon et al. [100] have produced AuNP by green synthesis of Brazilian red propolis (BRP) crude extract and its fractions (hexane, dichloromethane, and ethyl acetate) [100]. The reactions were optimized at pH 7. The AuNPs’ sizes produced ranged from 8 to 15 nm and were spherical, while different shapes were seen in the case of using ethyl acetate fraction, hexane and dichloromethane fractions, and BRP crude extract, respectively. All AuNPs displayed a crystalline structure with a face-centered cubic lattice and exhibited antimicrobial and cytotoxic activity. The method used represented low-cost and significant biological properties [100].

#### 4.2.3. Bee Pollen for Gold Nanoparticles (AuNPs) Synthesis 

AuNPs have been synthesized by Andean honey BP extract from Sangolqui, Ecuador. The biosynthesized NPs were analyzed by UV-vis spectroscopy, TEM, dynamic light scattering (DLS), Fourier-transform infrared spectroscopy (FTIR), and X-ray diffraction (XRD) spectroscopy to confirm their stability, morphology, optical properties, and purity. The results showed spherical and triangular AuNPs, with a size range of 7 to 42 nm. The stability was attributed to BP bioactive compounds such as proteins, fatty acids, polysaccharides, and polyphenols, as confirmed by FTIR. In addition, the study reported that green synthesis of AuNPs using BP is an effective, simple, and low-cost technique. Furthermore, the study demonstrated the effectiveness of the bee pollen-mediated AuNPs in the catalytic reduction of the organic pollutant 4-nitrophenol and its transformation to 4-aminophenol, and thus it is recommended for more studies, especially for the treatment of wastewater [54].

### 4.3. Bee Pollen for Magnesium Nanoparticle (MgNPs) Synthesis 

In a recent study, MgNPs were prepared using BP extract. The size of the obtained NPs was 36–40 nm, as characterized utilizing SEM, TEM, Fourier-transform infrared spectroscopy, energy-dispersive X-ray analysis, and X-ray diffractometer [53]. The UV-VIS absorption spectrum of BP@MgNPs was shown at 271 nm, and the surface charge that was obtained by biosynthesis was measured at a zeta potential of 12.9 mV using dynamic light scattering. The synthesized BP@MgNPs showed good antioxidant and antibacterial activity. Practically, DPPH radical scavenging activity was found to be 44% for BP@MgNPs, which can be explained by the presence of phenolic bioactive components in the aqueous extract of bee pollen. Additionally, these nanocomposite powders showed high antibacterial activity with minimum inhibitory concentration (MIC) values of 0.05 mg/mL for *S. aureus* and 0.25 mg/mL for *E. coli* [53]. 

### 4.4. Bee Bread for Iron Nanoparticles (FeNPs) Synthesis 

Bb sourced from the Bingöl region in Turkey was utilized for the development of iron nanoparticles, referred to as BB@FeNPs. These nanoparticles were characterized using various techniques such as SEM, X-ray diffraction spectrometry, Fourier transform infrared spectrophotometry, and ultraviolet-visible light spectrophotometry. Phenolic compounds, which are naturally occurring substances with both biological and nutritional benefits, are an ideal choice for the green production of NPs. They are particularly effective in stabilizing BB@FeNPs [115]. These nanoparticles have been introduced to the market as a safe alternative for sunscreens. Himalayan honey-loaded iron oxide nanoparticles (HHLIO-NPs) were synthesized and have an average particle size of 33–40 nm and are seen as needle-shaped porous structures. Additionally, they demonstrated scavenging activities and good antimicrobial properties against *E. coli* [116].

## 5. Biological Activities of Bee Products-Mediated Nanoparticles

Metal NPs are found to be potential therapeutic tools against various diseases, including cancer. The antioxidant, antifungal, antibacterial, and anticancer activities of green synthesized metal as well as metal oxide NPs using bee products have been demonstrated in various studies (Figure 7). 

The biosynthesized *Trigona* sp. honey-mediated CaONPs exhibited the highest antifungal effect against the anthracnose disease that is caused by *Colletotrichum brevisporum*. Additionally, it showed an anti-cytotoxicity effect against the human lung cell (MRC5) as well as the monkey kidney cell (VERO) [104]. SeNPs were synthesized using propolis extract utilizing ultra-sonication, self-assembling UV radiation, microwave irradiation, conventional heating, and hydrothermal methods [103]. These nanoparticles exhibited notable antioxidant activity, which varied in intensity. This could be explained by the ability of SeNPs to reduce oxidative stress via the interaction with free radicals and the blockage of adverse chain reactions [103].Furthermore, the antioxidant, antibacterial, and antifungal properties of AgNPs, produced using an aqueous extract of propolis, were identified. The antioxidant activity was assessed using the DPPH free radical scavenging assay and the lipoxygenase inhibition assay. The IC_50_ value indicated that the activity was close to 50% inhibition/scavenging [117]. Bb-AgNPs of Bb also exhibited antitumor activity against colon adenocarcinoma, ATCC HTB-37 cell line in a dose-dependent manner, in which the IC_50_ ranged between 24.58 to 67.91 µg/mL [91].

### 5.1. Anticancer

There is no doubt that cancer is a serious disease and one of the most common death causes in the world. According to the American Cancer Society, extracted from the National Center for Health Statistics for data collection, the number of new cancer cases was 1,958,310 and the number of cancer deaths was 609,820 cases only in the United States in 2023 [118]. Due to the seriousness of the matter, scientists are always looking for new and effective strategies to diagnose and treat cancer. Among the various strategies for cancer diagnosis and treatment, nanotechnology could provide promising solutions for the diagnosis and treatment of cancer [119]. Many studies have investigated the anticancer properties of honeybee products mediated by NPs. The ethanolic extract of propolis from stingless bees was used to mediate AgNPs and was proposed as a potential treatment for human lung carcinoma. The IC_50_ value was determined to be 38 µg/mL using the (3-[4,5-dimethylthiazol-2-yl]-2,5 diphenyl tetrazolium bromide) (MTT) assay. An increase in the concentration of AgNPs resulted in enhanced cell toxicity and reduced cell viability [40]. 

Al-Fakeh et al. in 2021 reported the anticancer activity of PdNPs synthesized by Saudi propolis against MCF-7, and an increase in cell toxicity and reduction of the cell viability with IC_50_ of 104.79 µg/mL was seen [41]. BV nano conjugates showed significant anticancer potential when loaded into the fungal chitosan, which was extracted from *Fusarium oxysporum*-grown mycelia. Another in vitro study evaluated the nano fungal chitosan (NFC), BV, and (BV/NFC) nanoconjugates against HeLa cervix carcinoma [67]. The nanoconjugates of BV with nano fungal chitosan exhibited anticancer properties with IC_50_ = 200 μg/mL. The early and late apoptosis and secondary necrosis markers were detected with DAPI (4′,6-diamidino-2-phenylindole) and acridine orange/propidium iodide fluorescent staining of HeLa cells [67]. In an in-vitro model, the anticancer activity of BV conjugated with nano chitosan from *Penaeus karatherus* (shrimp exoskeleton) was investigated against prostate cancer (PC3) and hepatocellular carcinoma (HEPG2). IC_50_ values of HepG2 were 20, 512, and 16.55 µg/mL for BV, chitosan NPs (CsNPs), and CsNPs + BV, respectively, while IC_50_ values of PC3 were 49.4, 410.2, and 36.08 µg/mL of BV, CsNPs, and CsNPs + BV, respectively. IC_50_ values of Vero were 156.8 and 311 µg/mL for BV and CsNPs + BV, respectively. The conjugate of NPs encapsulated in BV has a superior impact on HepG2 and PC3 cells relative to Vero cells. The inhibition of growth was dependent on the concentration and cell types, where at lower concentrations of BV, the expression of the P-gp pump in cancer cells was evident. A decrease in cytotoxic protein accumulation was evident when the nanoparticles with BV were directly delivered to the nuclei of cancer cells [68]. 

The AgNPs-G synthesized by aqueous extract of BP has been studied. AgNPs exhibited anticancer properties against MCF-7 and HepG2 compared to AgNPs-C (chemically synthesized AgNPs with sodium borohydride). The IC_50_ values of HepG2 and MCF7 were 96 ± 3.5 µg/mL and 47.6 ± 1.5 µg/mL of AgNPs-G, respectively, compared to IC_50_ of 214 ± 5 µg/mL and 310 ± 5.5 µg/mL related to AgNPs-C. AgNPs-G has higher antioxidant properties than AgNPs-C and commercially available ampicillin [120].

The Iranian propolis extract showed some potential anticancer activities, and its effect increased in combination with layered double hydroxide nanoparticles (LDH NPs), manifested in the upregulation of Bax pro-apoptotic gene expression and the downregulation of Bcl-2 anti-apoptotic gene expression [121].

### 5.2. Antibacterial

Nowadays, it has become an urgent matter to explore new antibacterial agents, as bacterial infection has become one of the main causes of mortality worldwide, especially in low-income countries. The misuse and overconsumption of antibiotics have led to the manifestation of multi-drug-resistant bacteria. Therefore, nanomaterials with antibacterial quality or that act as drug carriers have been widely developed as an alternative to fight off bacterial infections [122]. Several studies have studied the antibacterial activity of honeybee products-mediated nanoparticles.

Sidr honey (SH) and rhododendron honey (RH) were used for the green synthesis of AgNPs and have been evaluated for their antibacterial activity. AgNPs were tested against the Gram-positive bacteria, *S. aureus* and *Bacillus cereus,* and the Gram-negative bacteria, *Pseudomonas aeruginosa* and *E. coli*, using standard agar well diffusion. SH-AgNPs (1:1–5 mM) and RH-AgNPs (1:1–5 mM) showed significant antibacterial effects against the tested bacteria. Furthermore, they had better results against *S. aureus* than others, as the inhibition zone was 14 mm [26]. The study recommended synthesized NPs for enhancing the honey’s antibacterial activity [26].

The synthesis of ZnONPs using honey from India has been documented (as shown in Table 2). It was found that ZnO exhibited a stronger effect against *P. aeruginosa* compared to *E. coli*, with inhibition zones of 80% and 71% of the control treatment amikacin, respectively. Similarly, the inhibition zone diameter (in mm) of H-ZnONPs measurements for *Bacillus subtilis*, *S. aureus*, *E. coli*, and *P. aeruginosa* were 19, 16, 17, and 15, respectively. In comparison, the control treatment (amikacin) showed inhibition zone diameters of 29, 32, 24, and 17 for *B. subtilis*, *S. aureus*, *E. coli*, and *P. aeruginosa,* respectively [111].

In this regard, the antibacterial properties of Ag-Pt nanoparticles, mediated by propolis at concentrations of 25, 50, and 100 μg/mL, were studied against various bacteria, including *S. aureus*, *Klebsiella pneumoniae*, *E. coli*, *S. epidermidis*, *Serratia marcescens*, and *B. subtilis*. At a concentration of 100 μg/mL, bacterial viability ranged from 22.58% to 29.67% [88]. Additionally, a comparative analysis was conducted on the nanoparticle size and inhibition zones of AgNPs mediated by *Ziziphus spina*-christi and *Acacia gerrardii* honey, alongside amoxicillin, cefuroxime, and ciprofloxacin, against *S. aureus*, *E. coli*, and *P. aeruginosa*. The bactericidal activity of AgNPs, produced using honey from different sources, was also examined using the agar well diffusion method. AgNPs mediated by *Z. spina-christi* honey were found to be more effective than those mediated by *A. gerrardii* honey (Table 3) [76].

Furthermore, *Z. spina-christi* honey-mediated AgNPs revealed about 97%, 80%, and 88% inhibition rates for amoxicillin, cefuroxime, and ciprofloxacin, respectively. In the case of *E. coli*, *Z. spina-christi* honey-mediated AgNPs revealed 100%, 91%, and 67% inhibition rates for amoxicillin, cefuroxime, and ciprofloxacin, respectively. *A. gerrardii* honey-mediated AgNPs displayed inhibition rates of more than 85% for amoxicillin, 70% for cefuroxime and 70% for ciprofloxacin against *S. aureus* and 100% for amoxicillin, 76% for cefuroxime, and 55% for ciprofloxacin against *E. coli*. *A. gerrardii* honey-mediated AgNPs demonstrated good activity in comparison with amoxicillin and cefuroxime against *P. aeruginosa* [76]. The combination of the activity between the honey and the Ag ions may be the reason for the high activity of AgNPs as antibacterial agents [26]. The biosynthesized AgNPs fabricated by propolis extract from Romania showed antibacterial activity against *S. aureus* and *P. aeruginosa* with inhibition zone diameters of 10 ± 0.1 and 2 ± 0 mm, respectively [117].

The PdNPs synthesized by Saudi propolis extract were evaluated for antibacterial activity by the agar diffusion method against *S. aureus*, *E. coli*, *B. subtilis*, and *K. pneumonia*. The inhibition zones of *B. subtilis*, *E. coli*, and *K. pneumonia* (35 mm, 40 mm, 33 mm, respectively) were detected to be higher than *S. aureus* (23 mm). The study confirmed the significant antibacterial effect of PdNPs [41].

SeNPs have been studied for their antibacterial activity in which the SeNPs were prepared with propolis extract collected from five different Indian states and tested against *Salmonella typhi*, *E. coli*, *P. aeruginosa*, *B. Cereus*, *S. aureus*, and *S. mutans*. The Resazurin microtiter plate method was applied to detect the minimum inhibition concentrations of SeNPs (µg/mL) of *S. aureus*, *S. mutans*, *B. cereus* and *S. typhi*, which were 250, 500, 250, and 10,000, respectively [103].

## 6. Application of Bee Products-Mediated Nanoparticles

### 6.1. Catalysis Application

Nanotechnology, a rapidly evolving and expanding field, has found applications in a multitude of sectors [123,124]. NPs mediated by bee products have been utilized in a wide array of fields, one of which includes catalytic applications, as demonstrated in various studies [27]. In this study, AgNPs exhibited significant catalytic activity in reducing methylene blue dye. The degradation of methylene blue was analyzed using UV/Vis spectrophotometry and HPLC. After 72 h, the degradation of methylene blue reached 92%. Moreover, HPLC post-treatment spectra revealed the emergence of several new peaks [27]. Honey-mediated ZnONPs also have photocatalytic potential on methylene blue degradation, hence aiding in water treatment. The results showed that 86% of dye degradation was due to the honey-mediated NPs, compared to 60% by ZnO solely [111]. A study conducted by Venu, Ramulu, and their team explored the catalytic application of PtNPs mediated by honey sourced from South Korea. PtNPs exhibited impressive catalytic properties in the formation of antipyrilquinoneimine dye. Specifically, the PtNPs effectively catalyzed the reaction between aniline and 4-aminoantipyrine in an acidic aqueous medium. The formation of the dye was characterized using UV-visible spectroscopy [8].

The catalytic activity of Pt-AgNPs, synthesized using an aqueous extract of propolis from Turkey, has been examined in the context of hydrogen production. These Ag-Pt nanoparticles have been found to act as catalysts in the hydrolysis of sodium borohydride (NaBH_4_), a reaction that holds potential for hydrogen generation technology. The nanoparticles exhibited a high turnover factor (TOF) and a low activation energy of 1208.57 h^−1^ and 25.61 kJ/mol, respectively. This results in a high yield of hydrogen even at low temperatures [88].

AgNPs mediated by Romanian propolis exhibited significant photocatalytic activity related to the degradation of malachite green. When exposed to solar irradiation, 90% of the dye was degraded after 5 h. In contrast, only 30% degradation was observed in the dark, while the control degradation was at 5% [117]. The study highlighted the potential of using propolis-mediated AgNPs for the removal of organic dyes, offering a time-efficient, cost-effective, and eco-friendly solution [117]. AuNPs were also biosynthesized using Romanian propolis. These nanoparticles exhibited notable catalytic activities in the reduction reaction of 4-nitrophenol [42].

### 6.2. Food Industries

Ensuring food quality and safety remains an important challenge facing human health and welfare. The fast and incessant development of nanotechnology contributed to a breakthrough in food science, especially processing, packaging, and storage of food products [125]. *Kelulut* honey NPs are greatly recommended. This is according to a study that compared the respiration rates of papayas (*Carica papaya* L.) coated with *Kelulut* honey nanoparticles and uncoated (control) ones. The coating solution was kept in cold storage at 12 ± 1 °C for 21 days. A kinetic model was used to establish the changes in O_2_, CO_2_, and C_2_H_4_ production in papayas during the storage period. A closed system method was used to revise O_2_, CO_2_, and C_2_H_4_ after that represented by the Peleg model. The results demonstrated that the rate of respiration in the papayas during the storage significantly changed, with an increase in CO_2_ and decreases in O_2,_ and C_2_H_4,_ while the ascorbic acid level was recovered as well as the total phenolic content [39]. Furthermore, respiration rate was slowed in the treated papayas relative to the control. The study showed that Kelulut honey NPs are safe candidates that can be used in commercial post-harvest applications to extend the shelf life of papayas [39].

Chitosan nanoparticles (BCht) derived from *Apis mellifera* corpses, with a diameter of 151.9 nm, and SeNPs mediated by propolis, with a diameter of 11.2 nm, were assessed for their antimicrobial and preservative effects on catfish (*Clarias gariepinus*) fillets. The fillets were coated with various combinations of BCht, propolis, and SeNPs, forming an edible coating (EC) prepared from the tested composites. In contrast, the control group was coated with sterilized deionized water. After seven days of cold preservation, sensory, chemical, and microbial evaluations were conducted. The sensory parameters were represented by the appearance, color, odor, and overall quality of the samples. The edible coatings, made from the tested compounds and composites, resulted in the effective preservation of the catfish fillets’ color and appearance for seven days during cold storage. However, the control group showed unsatisfactory results in both appearance and texture. Equally interesting, the coated catfish fillets showed lower counts of psychrophilic bacteria, yeast, and molds. EC was found to be the most effective against all bacterial strains [126]. The antioxidant and antimicrobial abilities of ECs were explained by the polymetric nature of BCht particles, especially after combination with Pro and SeNPs and conjugation of chitosan NPs [127,128]. The study has recommended that ECs and their nanocomposites be employed in the bio-preservation of numerous seafoods [39].

The incorporation of a 10% ethanolic extract of propolis and 33% chitosan nanoparticles into edible film formulations has shown effective preservation of strawberries. This suggests that those films could be promising candidates for packaging applications to extend the shelf life of food products in the future. 

In another study, five formulations were used, including chitosan (C1), chitosan nanoparticles (C2), and the ethanolic extract of propolis (EEP) at varying concentrations (C3 = 10%, C4 = 20%, and C5 = 30%). These propolis-chitosan nanoparticle edible films were evaluated for their physicochemical properties and antimicrobial activity. The results indicated that the propolis-containing formulations (C3–C5) exhibited a strong inhibitory effect on *Listeria monocytogenes*, and C3 influenced *E. coli* after 24 h. Therefore, the addition of 10% EEP and 33% chitosan NPs into the edible film presents a feasible alternative for active packaging, particularly for strawberry preservation [127].

## 7. Conclusions

An accumulating body of data supports the use of the natural sources in the green synthesis of nanomaterials to avoid the disadvantages of chemical and physical methods and gain the great advantages of accessibility and bioavailability. Bee-derived substances are rich in unique metabolites and showcase a range of biological properties. Therefore, bee products are attractive choices for the green synthesis of nanomaterials. The bee products’ application in nanotechnology is in its infancy, with honey and propolis being the primary focus and other products like BP, Bb, RJ, BV, and beeswax receiving less attention. The environmentally friendly, cost-effective, and simple process of green synthesis using honeybee products for nanoparticle production offers great potential in diverse applications due to their anticancer, antibacterial, antifungal, and antioxidant activities. They were administrated as catalysts in the food industry and contributed to the safe preservation of food supplements.

To optimize these benefits and reduce the side effects of food additives, it is suggested to further study the potential use of honeybee products in green synthesis of NPs, to exploit their unique contents and minimize the adverse effects of synthetic chemicals. Furthermore, those NPs’ stability, safety, and mechanism in in vitro and in vivo studies should be discussed in detail. The promising results promote the inclusion of NPs mediated by bee products in clinical studies and drug discovery.

## Figures and Tables

**Figure 1 bioengineering-11-00829-f001:**
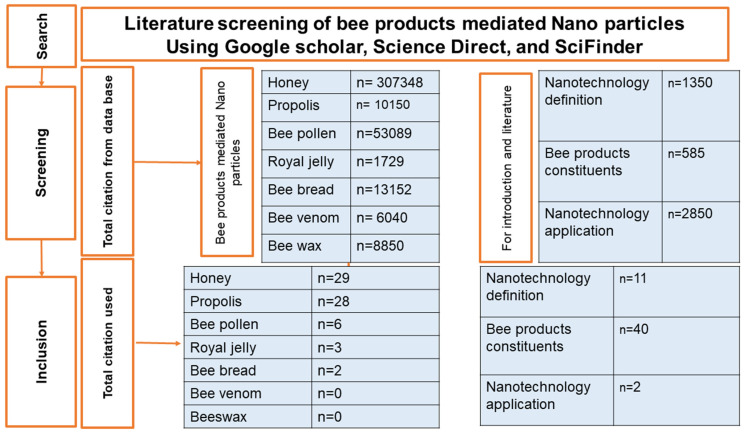
Flowchart of the systematic literature review strategy. n: Number of published papers.

**Figure 2 bioengineering-11-00829-f002:**
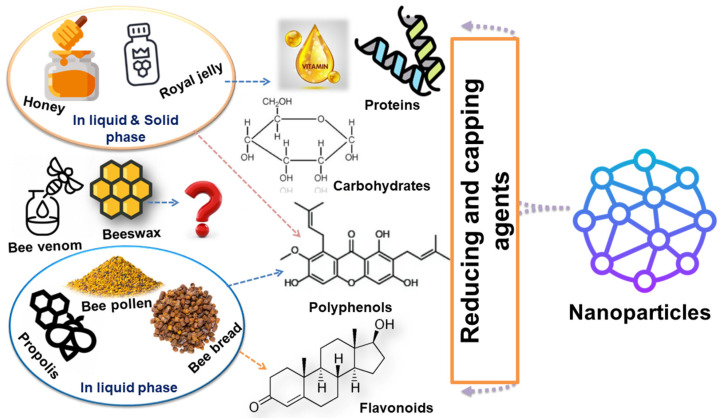
Deciphering the role of bee product metabolites in eco-friendly nanoparticle synthesis.

**Figure 3 bioengineering-11-00829-f003:**
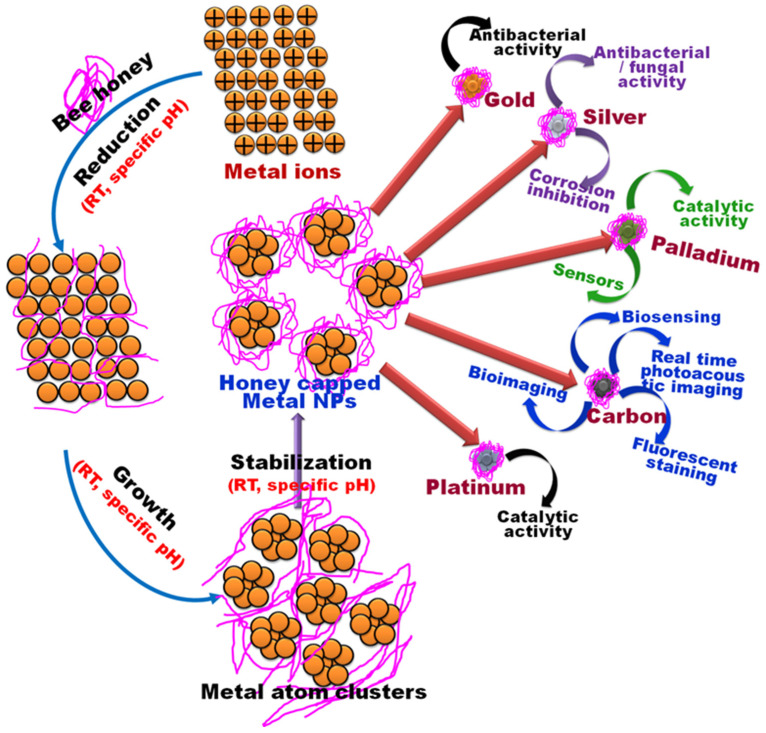
Honey mediated nanoparticles and their possible application (figure used with permission) [31].

**Figure 4 bioengineering-11-00829-f004:**
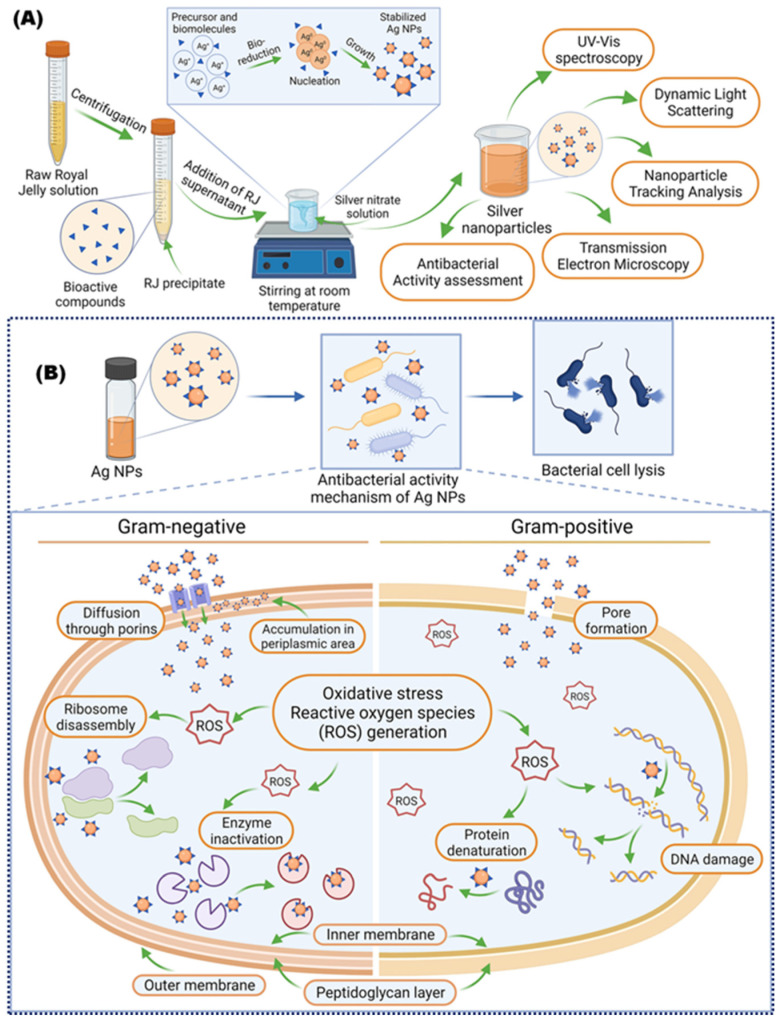
Diagrammatic illustration of the eco-friendly production of AgNPs using royal jelly (**A**), and the possible antibacterial action mechanism of AgNPs against both Gram-negative and Gram-positive bacteria (**B**) (Figure used with permission) [97].

**Figure 5 bioengineering-11-00829-f005:**
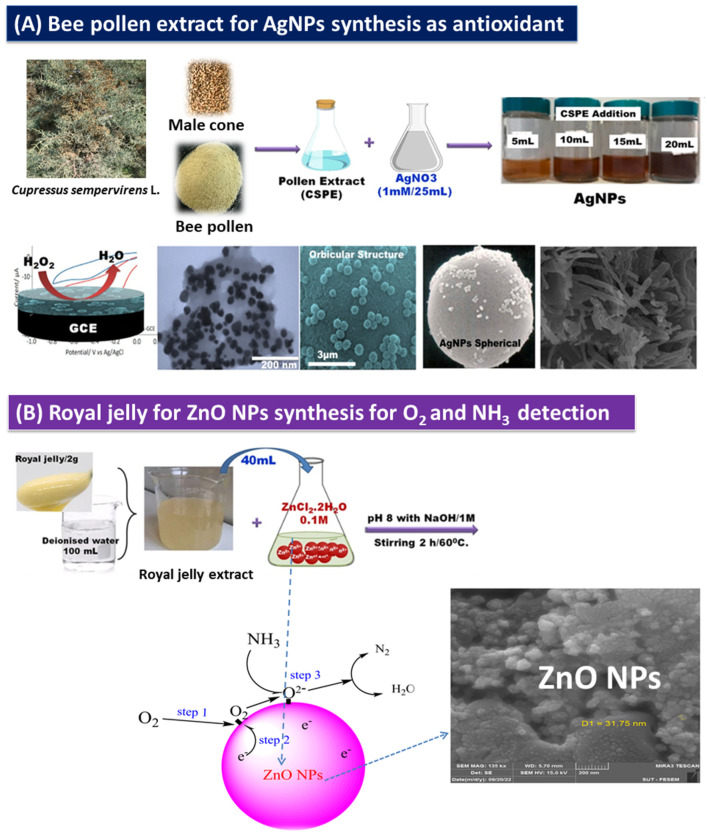
(**A**): Synthesis of AgNPs using *Cupressus sempervirens* pollen extract [55] and (**B**): ZnO synthesis using royal jelly (figure used with permission) [98].

**Figure 7 bioengineering-11-00829-f007:**
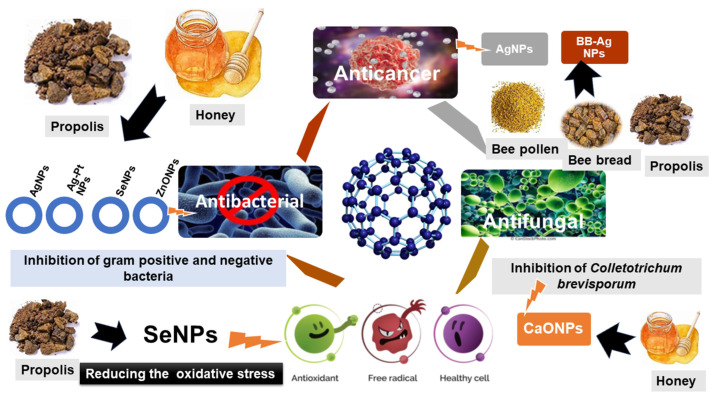
Highlighting the diverse biological impacts of nanoparticles derived from honeybee products.

**Table 1 bioengineering-11-00829-t001:** Synthesis of silver nanoparticles (AgNPs) bio fabricated using bee products.

Bee products Used/Reducing and Capping Agents	Morphology	NPs Size (nm)	Biological Activities and Application	References
**Utilizing honey for silver nanoparticles (AgNPs) synthesis**				
Honey/Honey	Spherical	9–22	Inhibitory effect on aflatoxins and ochratoxin	[69]
Honey/Glucose	Spherical	>20		[75]
Proteins, minerals, and polyphenols	Spherical with smooth edges	50–98	Antioxidant, cytotoxic, and antibacterial agent	[76]
Honey/Glucose, fructose, organic acids, vitamins, and minerals	Spherical	14.3 and 14.7	Antibacterial properties with inhibition zone of 14 mm against *Staphylococcus aureus*	[26]
Honey/Phenolic compounds, fructose, glucose, vitamins, and proteins	Spherical	42.7	Cytotoxic effect against L-929 cell line and antibacterial activity	[28]
Honey/Honey (black seed)	Spherical	25–70	Antibacterial activity	[77]
Honey/Fructose, glucose, and vitamin C	Spherical and monodispersed	5–25	Catalytic degradation of methylene blue	[27]
Honey/Alkane, ketone, alkene, nitro compounds, vinyl ether, and alkyl halide	Spherical	50–90	Anticancer, antimicrobial, immunomodulatory	[78]
Honey/Vinyl ether, alkene, and bromo compounds	Spherical	70–80	Antimicrobial, immunomodulatory, and cytotoxic activities	[79]
Honey/Fructose and proteins	Spherical	N.D.	Anticorrosion potentials for mild steel in acidic environments	[80]
Honey	Crystalline	11.1	-	[81]
**Utilizing propolis for AgNPs’ synthesis**				
Propolis/Polyphenols and flavonoids	Crystalline	91	Anticancer	[40]
Propolis/Quercetin and galangin	Spherical	16.5 ± 5.3	Antimicrobial activity	[82]
Propolis/Aliphatic –CH, –CH_2_ groups and carboxylate compounds in the propolis extract	N.D.	108.2	Antibacterial activity	[83]
Propolis/Amino acids, flavonoids, and phenolics	Spherical	8.7	Potent antimicrobial and wound-healing composites	[84]
Propolis/Polyol	N.D.	2–40	Antifungal, antibiofilm, and non-mutagenic properties	[85]
Propolis	Spherical	20	Antimicrobial activity	[86]
Propolis	Spherical	13–45	Immunomodulatory effect	[87]
Propolis	N.D.	5–15	Antibacterial effects and catalytic activity	[88]
**Utilizing bee pollen (BP) for AgNPs’ synthesis**				
BP	Spherical	11.5	Antioxidant activity	[55]
BP	Spherical	10–30	Anticancer activity and antioxidant activity	[89]
BP	Spherical	40–60 nm	Antidiabetic activity	[90]
**Utilizing bee bread (Bb) for AgNPs’ synthesis**				
Bb	Spherical	48.3–150.1	Antioxidant and antimicrobial activity	[91]
**Utilizing royal jelly (RJ) for AgNPs’ synthesis**				
RJ	Disk-like morphologies	4	Antimicrobial activity	[60]
RJ	Spherical	7.8–192.7 and 8.6–61.8	Antimicrobial activity	[61]

**Table 3 bioengineering-11-00829-t003:** Comparative analysis of nanoparticle size and inhibition zones of AgNPs mediated by *Ziziphus spina*-christi and *Acacia gerrardii* honey, alongside amoxicillin, cefuroxime, and ciprofloxacin, against *S. aureus*, *E. coli*, and *P. aeruginosa* [76].

Parameters	*Z. spina-christi* Honey-Mediated AgNPs	*A. gerrardii* Honey-Mediated AgNPs	*Amoxicillin*	*Cefuroxime*	*Ciprofloxacin*
Average size (nm)	50.5 ± 0.7	98.2 ± 0.9			
Inhibition zone of *Staphylococcus aureus* (mm)	19.4 ± 0.6	17.0 ± 0.1	20 ± 0.6	24 ± 0.9	22 ± 1.7
Inhibition zone of *Escherichia coli* (mm)	22.8 ± 1.2	19.0 ± 1.3	18 ± 1.4	25 ± 1.9	34 ± 1.3
Inhibition zone of *Pseudomonas aeruginosa* (mm)	21.0 ± 0.9	18.6 ± 0.8	-	-	35 ± 1.8

(-): No reported

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
