# Peer review of "Green Innovation and Synthesis of Honeybee Products-Mediated Nanoparticles: Potential Approaches and Wide Applications"

_bioengineering, 2024, doi:10.3390/bioengineering11080829_

Round 1
Reviewer 1 Report
Comments and Suggestions for Authors
The authors are suggested to cut down on the 'nanotechnology' part in the first paragraph and merge it with the second. Most of the readers are already familiar with the key attributes of nanotechnology, so there is no need to emphasize nanotechnology over the key attributes of honeybee products, making it an interesting candidate for nanoparticle synthesis.
2) Some of the numbers (with 2 decimal points) shown in the RHS column of Table 1 are confusing. Please replace the first decimal point with a comma.
3) Some formulas throughout the document need formatting,e.g., the numbers need to be converted to superscripts and subscripts
4) It would be worthwhile to mention existing challenges and future directions in the last paragraph prior to the Conclusions
4) Figure 5: What does the yellow labeling in part A depict?
Comments on the Quality of English LanguageThere are major mistakes in the spelling, sentence structure and punctuation throughout the document. Some of the sentences do not make any sense because of that,
Author Response
Response letter_ Reviewer 1: enclosed

Reviewer 2 Report
Comments and Suggestions for Authors
The authors present a review manuscript discussing a large variety of nanoparticle compositions and applications. The nanoparticles were prepared by bee corpse extracts, honey, pollen, propolis, bee wax and bee bread. The work is quite comprehensive and interesting, but a vast amount of claims are not referenced, references partially randomly placed, some sentences are not understandable, then there are writing and grammar errors. For this reason, I recommend major revision. To list the error list below was a lot of work, and the list is by all means not exhaustive, there are more errors.
11. „pol-len“ wrong spelling
22. „cata-lysts“ wrong spelling
33. „re-search“ wrong spelling
44. Page 2 line 55 reference missing
55. Page 2 line 58 reference missing
66. Page 2 line 60 reference missing
77. Page 2 line 61 reference missing
88. Page 2 line 63 reference missing
99. Page 2 line 66 reference missing for new types of drug delivery system I suggest PEM capsule reference1 and their automated production2 as proof that they can be upscaled. Btw. These are bottom up synthesis methods.
110. Page 2 line 69 reference missing
111. “Page 2 line 63 reference missing” point missing or writing error
112. Page 3 line 102: 3 references missing for the claim great attention
113. Page 3 line 107 reference missing.
114. Page 3 line 117 reference missing.
115. Page 4 line 149 spelling error “drug'seffectiveness”
116. Page 4 line 150 reference missing
117. „nowadays'smedical“
118. Page 5 line 163 reference missing
119. Page 5 line 165 reference missing
220. „,,“ wrong comma setting
221. „CoFe2O4“ sub-letter error
222. Continue 3.2. Page 5
223. Two times “caffeic acid” in one sentence
224. Page 5 line 194 reference missing
225. Page 5 line 210 reference missing
226. Page 6 line 214 reference missing
227. Page 6 line 217 reference missing
228. Page 6 line 221 reference missing
229. Page 6 line 223 reference missing
330. “ Bee pollen has been used for Ag, Mg, and Au NPs. “ sentence makes no sense, also no reference
331. Page 6 line 227 reference missing
332. Page 6 line 228 reference missing
333. Page 6 line 228 Since when does silver contain carbon, oxygen and nitrogen? The sentence makes no sense, also the authors did not state how the elements get inside and how the particles were in detail synthesized and they should state multi-element particles they contain more than just silver.
334. Page 6 line 231 for the claim significant efforts 3 references missing
335. Page 6 line 233 reference missing
336. Page 6 line 240 reference missing
337. Page 6 line 241 reference missing
338. Page 6 line 245 reference missing
339. Why are there two different font sizes in the text?
440. Page 6 line 248 reference missing
441. Page 6 line 254 reference missing
442. Page 6 line 258 reference missing
443. Page 7 line 265 reference missing
444. Page 7 line 270 reference missing
445. Page 7 line 274 reference missing
446. Page 7 line 276 reference missing
447. Page 7 line 282 reference missing
448. Page 7 line 289 reference missing
449. Why is there a link blue underlined in the middle of the manuscript? I thought links are properly cited?
550. Page 7 line 296 reference missing
551. Page 7 line 301 reference missing
552. Page 8 line 317 reference missing
553. Page 8 line 331 reference missing
554. Page 8 line 333 reference missing
555. Page 8 line 341 reference missing
556. Page 8 line 342 reference missing
557. Page 8 line 346 writing error
558. Page 8 line 317 reference missing
559. Page 11 line 381 reference missing
660. Page 11 line 393 reference missing
661. Page 11 line 404 reference missing
662. Page 11 line 405 writing error
663. Page 11 line 407 writing error
664. Page 12 line 414 reference missing
665. Page 12 line 424 reference missing
666. Page 12 line 429 reference missing
667. Figure 5 caption font is vastly different from other captions
668. Page 14 line 465 reference missing
669. Page 14 line 472 3+ is not superscript
770. Page 14 line 479 reference missing
771. Page 15 line 493 comma error
772. Page 15 line 502 writing error
773. Table 2 subscript errors
774. Page 17 line 511 reference missing
775. Page 18 line 527 reference missing
776. Page 18 line 529 reference missing
777. Page 18 line 542 reference missing
778. Page 18 line 550 reference missing
779. Page 18 line 550 symbol error
880. Page 18 line 554 font error
881. Page 18 line 555 reference missing
882. Page 18 line 560 reference missing
883. Page 18 line 562 font size error
884. Page 18 line 564 reference missing
885. Page 19 line 574 reference missing
886. Page 19 line 580 reference missing
887. Page 19 line 583 reference missing
888. Page 19 line 587 grammar and writing errors
889. Page 19 line 593 reference missing
990. Page 19 line 624 different citation style
991. Page 21 line 635 reference missing
992. Page 21 line 636 symbol error
993. Page 21 line 641 reference missing
994. Page 21 line 660 reference missing
995. Page 21 line 671 Symbol error
996. Page 21 line 674 reference missing
997. Page 21 line 680 reference missing
998. Page 22 line 687 reference missing
999. Page 22 line 703 reference missing
1100. Page 22 line 708 reference missing
1101. Page 23 line 721+722 different citation style
1102. Page 23 line 725 reference missing
1103. Page 23 line 731 reference missing
1104. Page 23 line 736 reference missing
1105. Page 23 line 743 reference missing
1106. Page 23 line 744 subscript error
1107. Page 23 line 749 reference missing
1108. Page 23 line 761 reference missing
1109. Page 23 line 762 writing error
1110. Page 24 line 763 writing error
1111. Page 24 line 771 writing error
1112. How can a coating be optimistic?
1113. Page 24 line 777 reference missing
1114. Page 24 line 795 reference missing
1115. Page 24 line 799 reference missing
1116. Page 24 line 806 Font error
References
(1) Sukhorukov, G. B.; Donath, E.; Lichtenfeld, H.; Knippel, E.; Knippel, M.; Budde, A.; Mohwald, H. Layer-by-Layer Self Assembly of Polyelectrolytes on Colloidal Particles. Colloids Surfaces A Physicochem. Eng. Asp. 1998, 137, 253–266. https://doi.org/10.1016/S0927-7757(98)00213-1.
(2) Li, W.; Gai, M.; Rutkowski, S.; He, W.; Meng, S.; Gorin, D.; Dai, L.; He, Q.; Frueh, J. An Automated Device for Layer-by-Layer Coating of Dispersed Superparamagnetic Nanoparticle Templates. Colloid J. 2018, 80 (6), 648–659. https://doi.org/10.1134/S1061933X18060078.
Comments on the Quality of English Language
many errors, the list is not complete, I was too tired to list all
Author Response
Response letter_ Reviewer 2: enclosed

Reviewer 3 Report
Comments and Suggestions for Authors
The manuscript submitted for evaluation is a review work and concerns the use of bee honey and its ingredients for the production of metal nanoparticles. In this work, the authors summarized and described the research results to date, but did not conduct a critical analysis of the problem. The problems that need to be solved in the future and the "real" possibilities of using such nanostructures, taking into account the extraordinary value of honey and its ingredients, have not been indicated. In addition, this manuscript requires linguistic correction and correction of typographical errors in words.
Comments on the Quality of English LanguageThis manuscript requires typological errors.
Author Response
Response letter_ Reviewer 3: enclosed

Round 2
Reviewer 1 Report
Comments and Suggestions for Authors
The manuscript reads well after the authors incorporated the suggested changes appropriately.
Reviewer 2 Report
Comments and Suggestions for Authors
I did not read everything again, but the manuscript was improved.